# A Single Nucleotide Mutation in Adenylate Cyclase Affects Vegetative Growth, Sclerotial Formation and Virulence of *Botrytis cinerea*

**DOI:** 10.3390/ijms21082912

**Published:** 2020-04-21

**Authors:** Xue Chen, Xiaohong Zhang, Pinkuan Zhu, Yiwen Wang, Yantao Na, Han Guo, Yunfei Cai, Haozhen Nie, Yina Jiang, Ling Xu

**Affiliations:** School of Life Sciences, East China Normal University, Shanghai 200241, China; xchenaylli@sina.com (X.C.); zxh849780842@163.com (X.Z.); pkzhu@bio.ecnu.edu.cn (P.Z.); ywwang@bio.ecnu.edu.cn (Y.W.); m18217165382@163.com (Y.N.); ggatecnu@163.com (H.G.); cai41777@126.com (Y.C.); sunny2010n@126.com (H.N.)

**Keywords:** *Botrytis cinerea*, whole-genome resequencing, adenylate cyclase, single-nucleotide mutation, sclerotial formation, conidiation, virulence

## Abstract

*Botrytis cinerea* is a pathogenic fungus that causes gray mold disease in a broad range of crops. The high intraspecific variability of *B. cinerea* makes control of this fungus very difficult. Here, we isolated a variant B05.10^M^ strain from wild-type B05.10. The B05.10^M^ strain showed serious defects in mycelial growth, spore and sclerotia production, and virulence. Using whole-genome resequencing and site-directed mutagenesis, a single nucleotide mutation in the adenylate cyclase (BAC) gene that results in an amino acid residue (from serine to proline, S1407P) was shown to be the cause of various defects in the B05.10^M^ strain. When we further investigated the effect of S1407 on BAC function, the S1407P mutation in *bac* showed decreased accumulation of intracellular cyclic AMP (cAMP), and the growth defect could be partially restored by exogenous cAMP, indicating that the S1407P mutation reduced the enzyme activity of BAC. Moreover, the S1407P mutation exhibited decreased spore germination rate and infection cushion formation, and increased sensitivity to cell wall stress, which closely related to fungal development and virulence. Taken together, our study indicates that the S1407 site of *bac* plays an important role in vegetative growth, sclerotial formation, conidiation and virulence in *B. cinerea*.

## 1. Introduction

*Botrytis cinerea* is a typical necrotrophic pathogen with a broad host range, affecting more than 1000 plant species [1]. It causes considerable pre- and postharvest economic damage, and has become the second most important plant pathogenic fungi [2]. *B. cinerea* can be propagated asexually through conidia and sclerotia and sexually through ascospores during its life cycle [3,4]. Moreover, it possesses a versatile arsenal of pathogenic factors, including cell wall-degrading enzymes, phytotoxins (botrydial and botcinic acid), phytohormones, reactive oxygen species, oxalic acid, and small RNAs, that are used to complete the infection process and finally result in the decay of plant tissues [1,3,5]. Unfortunately, no plant crop materials have been found to be completely resistant to *B. cinerea* [6,7]; thus, study of the pathogenic mechanisms of *B. cinerea* is of vital importance.

With recent improved molecular and genomic techniques, some conservative eukaryotic signal transduction pathways that regulate differentiation and pathogenicity, such as cAMP signaling, Ca^2+^-mediated signal transduction, the Ras superfamily and the mitogen-activated protein kinase (MAPK) pathway, have been deeply studied in *B. cinerea* [8]. In *B. cinerea*, key components of the cAMP pathway include *bac* (the gene encoding adenylate cyclase in *B. cinerea*), *bcpkaR* and *bcpka1*, knockout mutants of which are defective in growth and pathogenicity [9]. Adenylate cyclase (AC) is activated by the G-alpha subunit BCG1. Activated AC converts ATP to cAMP, which is recognized by its primary receptor, cAMP-dependent protein kinase (PKA) [10]. However, the regulatory mechanisms that act downstream of PKA and PKA-independent signaling cascades remain to be explored. BCG1 can also activate the Ca^2+^ signaling pathway through phospholipase C (BcPLC1), and knockdown of *bcplc1* in *B. cinerea* leads to defects in spore germination, sclerotium formation and virulence [11]. In the Ras superfamily, knockout of *bcras1* and *bcras2* attenuates mycelial growth and pathogenicity [9,12]. The MAPK signaling pathway transmits signals through the sequential activation of three interacting protein kinases. In *B. cinerea*, it has been reported that three MAPKs, *Bmp1*, *Bmp3,* and *Bcsak1*, are involved in growth and pathogenicity [13,14,15,16]. Furthermore, the filamentous fungi have evolved light signaling pathways which regulate growth, development and metabolism and allow them to adapt to changes in the environment. In *B. cinerea*, exposure to white light regulates the development of reproductive structures; the production of asexual conidia is induced by white light, whereas sclerotia, which are dormant structures, are formed in darkness [8]. It has been reported that the blue light receptor White collar complex and the downstream light-responsive transcription factors BcLTF1 and BcLTF2 are involved in these responses [17,18]. Moreover, some light-responsive regulatory genes such as *bcvel1* and *bclae1*, which encode members of the Velvet complex, have been shown to be involved in the regulation of growth, differentiation and pathogenicity in *B. cinerea* [19,20].

In addition to its wide host range and complex pathogenic strategies, the high intraspecific variability of *B. cinerea* increases the challenge of controlling this fungus. The intraspecific variation among *B. cinerea* strains causes them to show high variability in colony morphology, pathogenicity and drug resistance [21,22,23]. Single-nucleotide polymorphisms (SNPs), usually caused by errors in DNA replication and repair, are a major source of genomic variation [24]. For example, the two *B. cinerea* strains B05.10 and T4, which have been whole-genome sequenced, show significant differences in light-dependent differentiation, oxalic acid formation and pathogenicity [19]. Map-based cloning revealed that in the T4 strain, a single-nucleotide mutation in *bcvel1* that results in the formation of truncated proteins is related to phenotypic variation [19]. The natural variation of fungi is an evolutionary resource that promotes the adaptability of fungi to changing environments. At the same time, it provides a valuable resource for the study of genetic variation and can be useful in determining the genetic characteristics associated with the phenotype of interest.

During our culture of wild-type *B. cinerea* B05.10, we isolated a strain that showed defects in hyphal growth, fungal development and virulence. We named the natural variant strain B05.10^M^. Using whole-genome resequencing and site-directed mutagenesis, we confirmed a single mutation, S1407P, in the *bac* that results in development and pathogenic defects in B05.10^M^ strain. Since adenylate cyclase has been shown to be a core component of cyclic AMP (cAMP) signaling, which controls a range of physiological processes in fungi [25,26], we performed physiological and biochemical assays to further investigate the effect of S1407 on BAC function. The results showed that S1407 site on BAC plays an important role in the accumulation of intracellular cAMP, spore germination rate, infection cushion formation, and even in cell wall integrity. The novel findings regarding the identification of a deterministic functional site (S1407) in BAC provide new insight regarding the function of BAC in fungal pathogenesis.

## 2. Results

### 2.1. Phenotypes of B05.10 and B05.10^M^ are Significantly Different in Vegetative Growth, Sclerotial Formation, Conidiation, and Virulence

B05.10 is used as a standard wild-type strain in the study of *B. cinerea*. Its major growth phenotype includes a light response in which the production of conidia is promoted in light, and it exhibits the formation of sclerotia in darkness (Figure 1A). In a preliminary study in our laboratory, a variant strain (later named B05.10^M^) originating from B05.10 was identified. The growth rate and biomass accumulation of B05.10^M^ was significantly reduced compared with B05.10 (Appendix A). After 3 days of incubation, the mean colony diameters of B05.10 and B05.10^M^ were 78.19+/−0.46 mm and 46.22+/−0.93 mm, respectively (Figure 1A,B; Appendix A). Notably, after 10 days incubation, B05.10^M^ produced conidia in both light and darkness but no sclerotia (Figure 1A,C,D), but the number of conidia in light was significantly lower than that of B05.10 (Figure 1C; Appendix A).

Virulence assays showed that the pathogenicity of B05.10^M^ to three different hosts (tomato leaves, grape and apple fruits) was significantly decreased (Figure 2A). The infected area produced by B05.10^M^ on those three hosts was reduced by more than 50% compared with B05.10 (Figure 2B). To investigate the mechanism of reduced virulence in B05.10^M^, we performed assays that measured spore germination and infection cushion formation. The results showed that the spore germination rate of B05.10^M^ was significantly lower at 4 h and 6 h than that of B05.10 (Figure 2C,D). At the same time, approximately half (45.47+/−3.08%) of the B05.10 colonies had formed infection cushions 12 h after inoculation, but no similar infection structures were observed in B05.10^M^ until 24 h after inoculation (Figure 2C,E). This suggests that the reduced virulence of B05.10^M^ compared to that of B05.10 might be due to an effect of the mutation on the initial stages of the infection process.

### 2.2. Whole-Genome Resequencing Analysis of B05.10^M^ and B05.10

To investigate the genotypic differences between B05.10^M^ and B05.10, three independent whole-genome resequencings of the two strains were performed using the Illumina Hiseq 2000 sequencing system. All datasets have been deposited to the GenBank database under accession number PRJNA595649 (https://dataview.ncbi.nlm.nih.gov/object/PRJNA595649?reviewer=tll6mr4stddkbj2tiac2qb2u5r).

Comparing with the reference genome of B05.10, there were approximately 800-900 SNPs in each set of resequencing data obtained from B05.10^M^. The SNPs located in exons and causing non-synonymous mutations were analyzed emphatically (Figure 3A,B; Appendix A). Among these SNPs, 18 non-synonymous SNPs were found in the data from three independent resequencings of B05.10^M^, and five of these SNPs were common to the three repeats (Figure 3C,D). Based on major functional domain analysis of the protein and Sanger sequencing verification, we speculated that a single-nucleotide mutation (T to C) at locus 4,646 of the *bac* (Bcin15g02590.1) that results in a change from serine (S) to proline (P) at position 1407 in adenylate cyclase (BAC) (Figure 4A,C), and may be related to the phenotypic variation observed in B05.10^M^ strain.

### 2.3. The S1407P Mutation in the Adenylate Cyclase Encoding Gene Bac Confers Serious Defects in Development and Virulence 

To verify the effect of the S1407P mutation of *bac*, we performed site-directed mutagenesis of the *bac* in B05.10 and B05.10^M^ using a homologous recombination strategy to generate in situ point mutation strain and complementary strain (Figure 4B). The *bac* with an S1407P mutation was introduced into B05.10 to generate in situ point mutation strain B05.10:*bac*^S1407P^, and *bac* with a P1407S mutation was introduced into the B05.10^M^ strain to generate *bac* complementary strain B05.10^M^:*bac*^P1407S^ (Figure 4B,C; Appendix A). 

The B05.10:*bac*^S1407P^ strains showed growth and pathogenicity phenotypes similar to that of the B05.10^M^ strain (Figure 1 and Figure 2; Appendix A). In brief, the growth rate and biomass accumulation of B05.10:*bac*^S1407P^ in both light and darkness were significantly lower than those of B05.10 (Figure 1A,B; Appendix A). In addition, B05.10:*bac*^S1407P^ produced conidia under both dark and light conditions but did not form sclerotia (Figure 1A,C,D). Moreover, the virulence of B05.10:*bac*^S1407P^ on three hosts was significantly reduced due to its decreased spore germination rate and a reduced rate of infection structure formation (Figure 2). In contrast, the phenotype of B05.10^M^:*bac*^P1407S^ with respect to growth rate, sclerotial formation, conidiation and virulence was similar to that of B05.10 (Figure 1 and Figure 2; Appendix A). These results indicate that the S1407P mutation in the *bac* caused severe defects in development and virulence in *B.cinerea*.

### 2.4. The S1407P Mutation in BAC Affects its Enzymatic Activity

In vivo, adenylate cyclase (AC) has been shown to be an important downstream effector of G-alpha protein. Its product, cAMP, acts as a second messenger to activate cAMP-dependent PKA [10]. Functional domain prediction and alignment showed that the point mutation from serine (S) to proline (P) at position 1407 of B05.10^M^ was located in the PP2Cc--type phosphatase domain of BAC, which was very conservative in different fungi (Appendix A), suggesting that the S1407 may play a critical role in the function of BAC.

To further understand the effect of the S1407P mutation on BAC function, the intracellular cAMP levels of B05.10, B05.10^M^, B05.10:*bac*^S1407P^, and B05.10^M^:*bac*^P1407S^ were determined. The results showed that the cAMP contents of the S1407P point mutation strain B05.10:*bac*^S1407P^ and B05.10^M^ were significantly decreased compared to B05.10, while the cAMP content of B05.10^M^:*bac*^P1407S^ showed no significant change (Figure 5A). 

Previous research showed that exogenous application of cAMP can partially restore the growth rate caused by functional defects in BAC [27,28,29]. Therefore, we further explored whether exogenous cAMP addition could reverse the growth defects exhibited by the S1407 point mutation strains (Figure 5B). As shown in Figure 5B, exogenous addition of cAMP suppressed the growth of B05.10 and B05.10^M^:*bac*^P1407S^ in a dosage-dependent manner. In contrast, exogenous addition of cAMP at 0.2 mM and 0.4 mM significantly increased the growth of B05.10^M^ and B05.10:*bac*^S1407P^. Taken together, the results indicate that the S1407P point mutation in BAC affected the activity of the enzyme and was very important for synthesizing intracellular cAMP.

### 2.5. The S1407P Mutation in Bac Alters the Fungal Cell Wall Integrity

Besides growth defects associated with the *bac* point mutation, we explored the relationship of the S1407 site in BAC to cell wall integrity, a parameter that is closely related to fungal development and virulence [15]. The sensitivity of point mutants to cell wall-damaging agents and degrading enzymes was determined (Figure 6). Compared with B05.10 and B05.10^M^:*bac*^P1407S^, the two *bac* S1407P point mutants (B05.10^M^ and B05.10:*bac*^S1407P^) showed enhanced sensitivity to Congo red (Figure 6A,B). In addition, the mycelia of the two *bac* S1407P point mutants were more easily digested to generate protoplasts than those of B05.10 and B05.10^M^:*bac*^P1407S^, after treatment with lysing enzymes at 23 °C for 2 h (Figure 6C,D). These results together confirmed that the cell wall integrity of *bac* S1407P point mutants were defective.

## 3. Discussion

The genetic diversity of natural populations of *B. cinerea* causes them to exhibit extensive phenotypic variation and leads to great differences in reproductive patterns, pathogenicity and drug resistance [30,31,32]. During saprotrophic culture, fungi may also exhibit natural phenotypic variation due to various environmental factors [33]. With the development of high-throughput sequencing technology, genome-wide resequencing has been widely used to reveal the genetic diversity of species and to identify novel candidate genes associated with phenotypes in different organisms [34]. In our study, a phenotypic variant strain, B05.10^M^, was derived from the sequenced wild-type strain B05.10 of *B. cinerea* during culture; it showed significant differences in mycelial growth, reproductive differentiation and virulence compared to B05.10 (Figure 1 and Figure 2; Appendix A). Based on the results obtained using whole-genome resequencing, we speculated that the 4646-locus single-nucleotide mutation (T mutated to C) of the adenylate cyclase-encoding gene *bac* caused the replacement of serine (S) to proline (P) at position 1407 of adenylate cyclase and resulted in phenotypic variation of B05.10^M^ (Figure 4A).

There has been some research on the function of SNP in *B. cinerea*. For example, most SNPs in *mbc1* gene encoding *β-tubulin* have been shown to be closely related to fungicide resistance [35,36]. Some SNPs, such as a single nucleotide mutation in *bcvel1* encoding VELVET protein, lead to the premature formation of stop codon, thus leading to the production of truncated proteins and the change of the developmental and pathogenic phenotypes [19,37]. In the study, we confirmed the single nucleotide mutation in *bac* at the genetic level through the acquisition and phenotypic analysis of *bac* in situ point mutants. To generate point mutants, an expression vector carrying a target gene with a specific point mutation is usually transformed into a deletion mutant of the target gene, thus generating the point mutant [38,39]. The method is similar to the usual complementary strategy. In this study, however, the *bac* point mutation site was introduced in situ at the locus of the *bac* based on a homologous recombination strategy; the resulting mutants are often more stable, and the method eliminates the interference caused by other genes (Figure 4B,C; Appendix A).

Adenylate cyclase (AC) has been shown to be an important downstream effector of G-alpha protein. Its product, cAMP, acts as a second messenger to activate cAMP-dependent PKA (protein kinase A), which phosphorylates downstream target proteins and affects cell metabolism [10]. To date, several AC-encoding genes of phytopathogenic fungi have been reported. The function of AC in fungi is relatively conserved; it is mainly involved in the organism’s growth and development, reproductive differentiation, virulence and stress tolerance [25,28,40,41]. For example, in *M. oryzae*, knockout of *MAC1* resulted in decreased vegetative growth and spore yield, defective appressorium formation and decreased pathogenicity [42]. In *B. cinerea*, the deletion of *bac* led to defects in sclerotia formation, decreased mycelial growth, sporulation and virulence. Furthermore, BAC relies on PKA to regulate virulence, but the regulation of growth rate, conidiation and sclerotia formation does not depend on PKA [9]. In the study, the results showed that the S1407P mutation in *bac* caused severe defects in development and virulence (Figure 1 and Figure 2), which is similar to the phenotype of *bac* knockout mutant [9], indicating that the S1407 site may play a crucial role in the function of BAC.

ACs have been reported to play a regulatory role in stress tolerance in fungi [43,44]. For example, in *Colletotrichum higginsianum*, Δ*ChAC* showed increased sensitivity to the cell wall-damaging agent Congo Red (CR), which breaks the connection of chitin microfibrils to β-glucan [26,45]. Here, we found that the S1407P point mutants B05.10:*bac*^S1407P^ and B05.10^M^ showed enhanced sensitivity to Congo red and generated more protoplasts when treated with lysing enzymes (Figure 6), suggesting that the S1407P mutation in *bac* may affect cell wall integrity. In addition, the S1407P mutation in *bac* resulted in decreased intracellular cAMP levels, however, exogenous addition of cAMP partially restored the growth of point mutants B05.10:*bac*^S1407P^ and B05.10^M^ (Figure 5). This results and trend is similar to that reported in *bac* mutant [29], indicating that S1407 is very important for BAC. Moreover, in *Aspergillus flavus* and *Penicillium digitatum*, the growth rate of *acyA* and *Pdac1* deletion mutants were also only slightly restored by the addition of exogenous cAMP [27,28]. AC has multiple functional domains, which may have other functions in addition to producing cAMP. The typical AC amino acid sequences of plant pathogenic fungi contained G-alpha binding domains, Ras association domains, PP2Cc-type phosphatase domains and AC catalytic enzyme domains. In the study, the mutant amino acid S1407 is conserved in fungi and located in the BAC phosphatase domain (Appendix A). According to the amino acid characteristics, serine is a polar amino acid, while proline is a nonpolar amino acid. Due to the R group of proline forms a loop with the amino group, it is easy to affect the formation of the α-helix of the secondary structure of the protein, making α-helix turn into a β - fold, which may affect the conformation of protein. We further used the SWISS-3d web to predict the protein structures of two BAC (Appendix A); it was found that the S1407P mutation protein BAC^S1407P^ and wild type BAC^S1407^ have different protein conformations. Therefore, we speculated that the S1407P mutation may affect the protein conformation of BAC, resulting in decreased enzyme activity, thus affect the development and pathogenicity.

In conclusion, in this study, a single-nucleotide mutation in the *bac* encoding AC of *Botrytis cinerea* was identified using whole-genome resequencing and site-directed mutagenesis approaches. Our results indicated that the S1407P mutation in *bac* played important roles in vegetative growth, sclerotia formation, conidiation and virulence. To our knowledge, this is the first report in which a key functional site in fungal AC was identified using a genetic approach and our findings provide important insights into reveal the development and pathogenic mechanism of *B. cinerea*.

## 4. Materials and Methods

### 4.1. Fungal Strain and Culture Conditions

The wild-type *B. cinerea* strain B05.10 was used in the present experiments [46]. The strain obtained from natural variation of the wild type during culture was named B05.10^M^. The strains are generally cultured on potato dextrose agar (PDA) medium for vegetative growth. In this study, however, a complete medium (CM medium) (30 g sucrose, 1 g KH_2_PO_4_, 0.5 g MgSO_4_·7H_2_O, 0.5 g KCl, 2 g NaNO_3_, 2.5 g N-Z Amine, 1 g yeast extract, 10 mL vitamin stock solution, 0.2 mL trace element solution in 1 L water plus 20 g agar for solid medium) was used to avoid the unknown effects of the components of PDA medium. All culture experiments were conducted at 23 °C.

### 4.2. Generation of bac Site-Directed Mutants

The *bac* in situ point mutants were generated by transforming a homologous recombination fragment constructed by fusion PCR according to a previously reported method [47] into the B05.10 and B05.10^M^ strain. The strategy for the *bac* site-directed mutants’ construction is shown in Figure 4B.

Briefly, genomic DNA of *B. cinerea* (B05.10 and B05.10^M^) was used as a template to PCR amplify both the fragment from approximately 1 kb upstream of the mutation site to the terminator of the *bac* (5′-region) and the 3′-region (including 1 kb downstream of the *bac* terminator) using the specified primer pairs P1/P2 and P5/P6, respectively. The plasmid pNAN-OGG containing a nourseothricin resistance cassette (*nat*) were used as templates to amplify the *nat* fragments with the primers P3/P4. The linkage products of the 5′-region, the 3′-region and the *nat* constitute the transformation fragment.

The transformation fragment was cloned into the vector pENTR™/SD/D-TOPO^®^ and used to transform *Escherichia coli* (DH5α) competent cells. After confirming the correct insertion of the fragment by sequencing, the plasmid was linearized using a restriction enzyme, transformed into the B05.10 or B05.10^M^ strain via protoplast transformation, and selected by nourseothricin stress [48]. The site-directed mutant strains were identified by diagnostic PCR of specific fragments and gene sequencing. The primer sequences are shown in Appendix A.

### 4.3. Analysis of Growth and Development Phenotype

To compare the growth of the wild-type and mutant strains, mycelial plugs (5 mm diameter) were excised from the edges of 3-day-old colonies, transferred to the center of fresh CM solid medium and incubated in the dark or light. Three parallel samples of each strain were examined for colony growth. Each experiment was repeated three times. After 10 days of incubation, the conidia produced were collected from each sample using 0.1% Tween solution, and the spores were counted. The number of sclerotia was also recorded. In addition, the mycelial plugs were inoculated onto CM solid medium with cellophane. After 3 and 5 days of culture, the mycelial samples were collected and dried at 65 °C to constant weight, and the weight was recorded.

To observe conidial germination and the formation of an infection cushion, spores harvested from each strain were incubated in liquid GB medium (3.05 g Gamborg B5, 3.6 g glucose in 1 L water) on glass slides placed in a humid box. After 4 h and 6 h of incubation, germination rates were determined. The germinated spores were stained with lactophenol cotton blue to make it possible to observe and count the formation of infection structures after incubation for 12 h and 24 h.

For the addition experiment of exogenous cAMP, the strains were inoculated on CM medium with different cAMP (Macklin, China) concentrations. After 3 days of incubation, the colony diameters were measured. To determine the sensitivity to cell wall stress, fungal strains were inoculated onto CM medium with 0.8 mg/mL Congo Red (Sangon, China) for 3 days and colony diameters were measured.

### 4.4. Virulence Assay

Leaves of 4-week-old tomato (*Solanum lycopersicum*) plants and healthy mature fruits of apple (*Malus pumila Mill.*) and grape (*Vitis vinifera*) of uniform size were used in the inoculation assay. Prior to inoculation, the fruits were rinsed with sterile water, submerged in 70% alcohol for 30 s, and soaked in sterile water for 2 min. For inoculation, 10-µL droplets of conidial suspensions (1 × 10^6^/mL GB) were dropped onto needle-injured sites on the fruit. Tomato leaves were inoculated with conidial suspensions on the leaf surface. The inoculated fruits and leaves were placed in a container at 90% relative humidity. The area of the lesions formed at the inoculation site was measured at the appropriate time.

### 4.5. Genome Resequencing Analysis

High-quality genomic DNA extracted from strains B05.10 and B05.10^M^ was used in genome sequencing. Sequence alignment of the sequenced reads and the reference genome sequence of B05.10 (http://fungi.ensembl.org/Botrytis_cinerea/Info/Index) was performed by BWA software, removing sequencing reads generated by PCR-duplication using Picards-tools and calculating the sequencing depth and coverage relative to the reference genome. Single-nucleotide polymorphisms (SNPs) and small insertion–deletion sites (Small InDel) were detected using VarScan software, sites with lower sequencing depth and lower quality values were filtered, and detection of structural variation (SV) of samples was performed using classic BreakDancer software. Annotation information for SNP, InDel, and SV was obtained by Annovar software and reference genome gff information. Finally, gene family enrichment analysis of SNP-related genes was performed based on interpro annotation information of SNP-corresponding genes. In this study, we mainly focused on single-nucleotide mutation sites (SNPs). The SNPs of B05.10 were excluded, and the SNPs shared by the three independent resequencing results of B05.10^M^ were focused on. The amino acid variation caused by these SNPs and the main functional domains of the encoded proteins were analyzed; Sanger sequencing of the candidate mutation genes was then performed for verification.

### 4.6. Mycelia Lysis Assay

Spores harvested from wild type strain and point mutants (the initial concentration of conidia was 10^8^ /mL) on CM were incubated in liquid ME medium (4 g Glucose, 4 g yeast extract, 10 g Maltose extract in 1 L, PH 5.5) at 180 rpm for 16 h at 23 °C. The young mycelia were then collected and washed with KC buffer (44.9 g KCl, 5.6 g CaCl_2_ in 1 L), followed by the treatment with cell wall-degrading enzyme (Sigma-Aldrich, USA) for 2 h at 100 rpm at 23 °C to produce the protoplasts. The number of protoplasts were counted.

### 4.7. Measurement of Intracellular cAMP Content

Three-day-old mycelium samples were frozen in liquid nitrogen and lyophilized. The dried samples were ground to a fine powder in liquid nitrogen using a Retsch mill. For each sample, 50 mg of the powdered sample was resuspended in 250 µL ice-cold 5% trichloroacetic acid (TCA) and incubated on ice for 10 min. After centrifugation at 4000 rpm for 15 min at 4 °C, the supernatant was collected and diluted for the assay. Intracellular cAMP content was measured using a highly sensitive ELISA kit (Biosamite, China) according to the manufacturer’s instructions.

### 4.8. Statistical Analysis

All experiments in this study were repeated three times. The data obtained were analyzed with ANOVA followed by Duncan’s multiple range tests (*p* < 0.05) for means comparison using SPSS 17.0. Values are the mean ± SE of measurements.

## Figures and Tables

**Figure 1 ijms-21-02912-f001:**
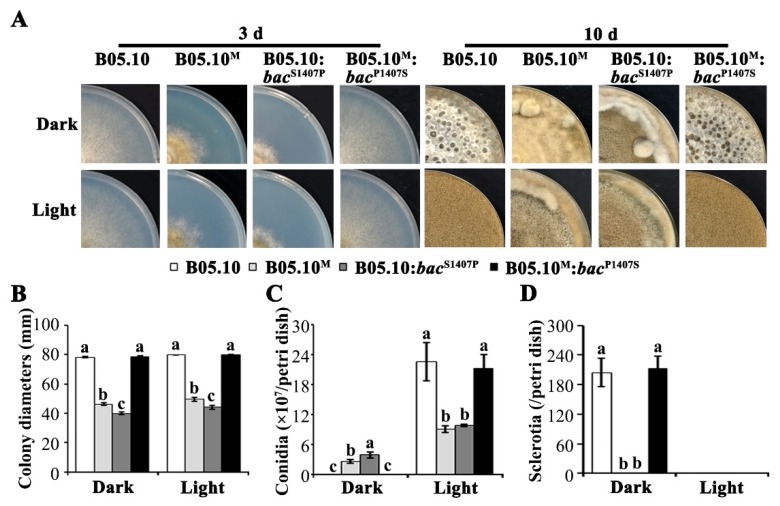
The S1407P mutation in *bac* resulted in defects in mycelial growth rate, sclerotial formation and conidiation. (**A**) Colony morphology of B05.10, B05.10^M^, B05.10:*bac*^S1407P^, and B05.10^M^:*bac*^P1407S^ in dark and light on CM after incubation for 3 and 10 days. (**B**) Colony diameters for the above strains in dark and light after incubation for 3 days. (**C**,**D**) Number of conidia (**C**) and sclerotia (**D**) per Petri dish formed on CM after 10 days of incubation, respectively. Different letters on the columns indicate significant differences (*p* < 0.05).

**Figure 2 ijms-21-02912-f002:**
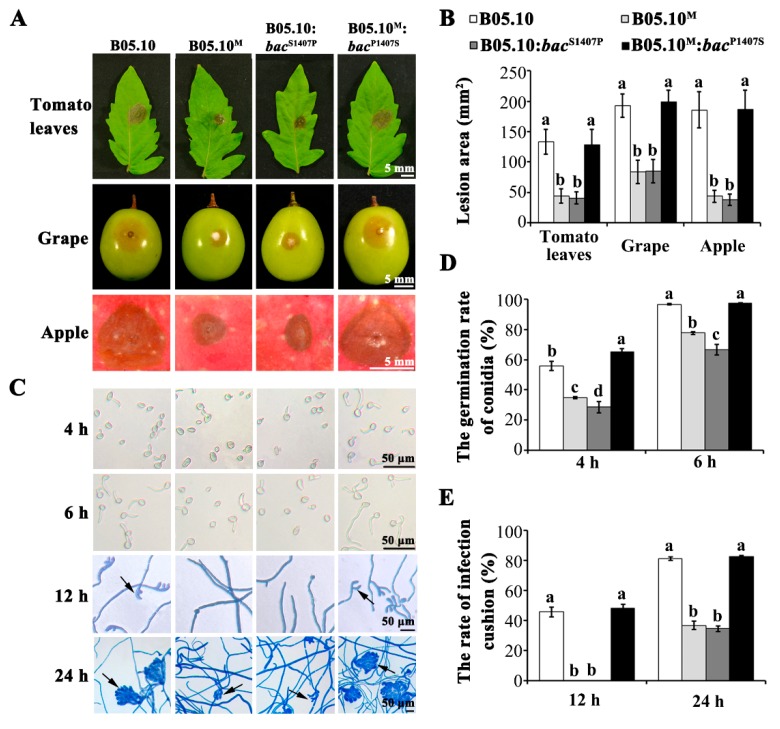
The single-nucleotide mutation in *bac* led to reduced virulence and decreased rates of spore germination and infection cushion formation. (**A**) Lesion formation on tomato leaves, grape and apple fruits inoculated with conidial suspensions (10^6^/mL) of B05.10, B05.10^M^, B05.10:*bac*^S1407P^, and B05.10^M^:*bac*^P1407S^. Disease symptoms on tomato leaves, grape and apple fruits were photographed at 48, 72, and 96 h after inoculation, respectively. Bar = 5 mm. (**B**) Area of lesions formed on tomato leaves, grape and apple fruits inoculated with four test fungi. (**C**) Spore germination and infection cushion formation. The infection cushions were stained with lactophenol cotton blue. Bar = 50 μm. (**D**,**E**) Rates of spore germination (**D**) and infection cushion formation (**E**) of each strain shown in (C). Different letters on the columns in (**B**), (**D**) and (**E**) indicate significant differences (*p* < 0.05); the same-colored columns in these graphs represent the same fungal strains shown in (**B**).

**Figure 3 ijms-21-02912-f003:**
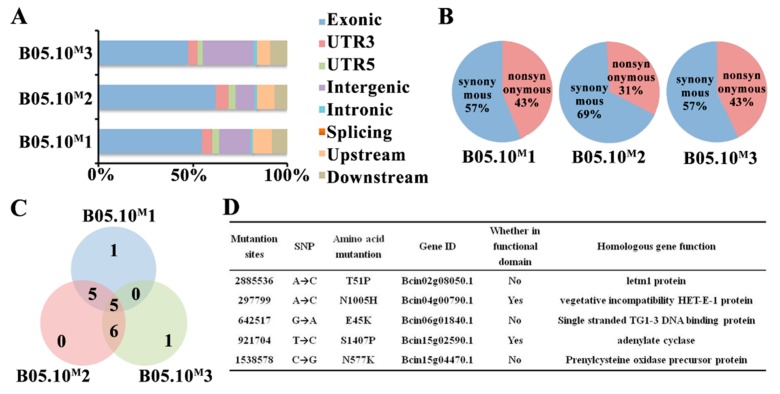
Whole genome resequencing analysis of B05.10^M^. (**A**) Distribution of single-nucleotide polymorphisms (SNPs) in the genome. B05.10^M^ 1-3 show data from three independent resequencings of B05.10^M^. (**B**) SNP-induced amino acid variation in the exon region. (**C**) Statistical analysis of nonsynonymous mutation in B05.10^M^ triple-sequencing data. (**D**) Analysis of 5 common missense mutation sites.

**Figure 4 ijms-21-02912-f004:**
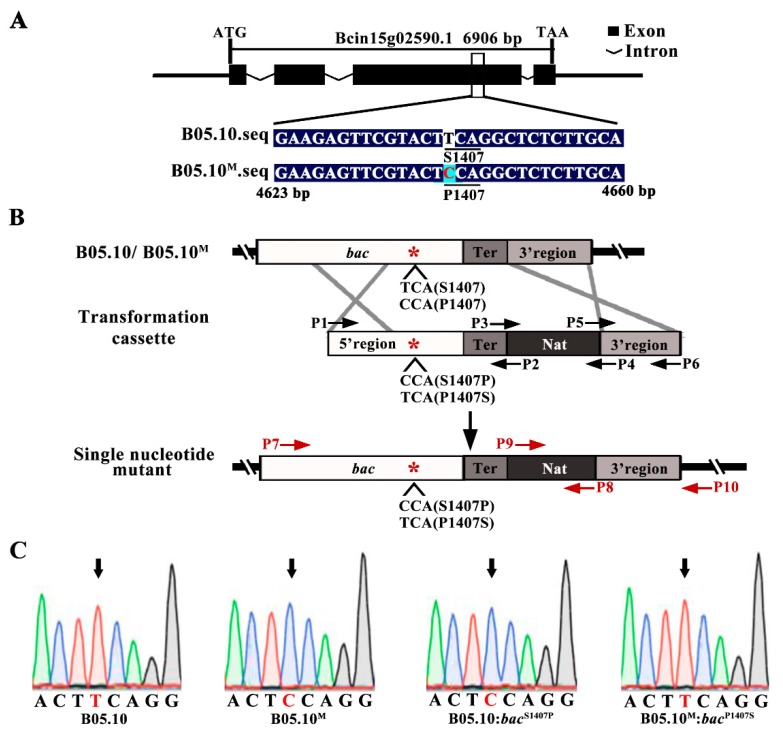
Structural diagram of the *bac* and construction of B05.10:*bac*^S1407P^ and B05.10^M^:*bac*^P1407S^ strains. (**A**) Schematic representation of S1407P mutation in *bac*. Black boxes and broken lines indicate the location of exons and introns in the *bac*. Comparative alignment of nucleotide sequences of the target site (the region between 4623 and 4660 nucleotide positions) of B05.10 and B05.10^M^ were shown at the bottom. (**B**) Schematic representation of the in situ point mutation strategy to construct B05.10:*bac*^S1407P^ and B05.10^M^:*bac*^P1407S^ strains. The asterisk (*), Ter, and Nat indicate the mutation site, terminator site, and nourseothricin resistance cassette, respectively. (**C**) Verification of single-nucleotide mutation sites (between nucleotide positions 4643 and 4650) by PCR-based Sanger sequencing. The mutation site is indicated by an arrow.

**Figure 5 ijms-21-02912-f005:**
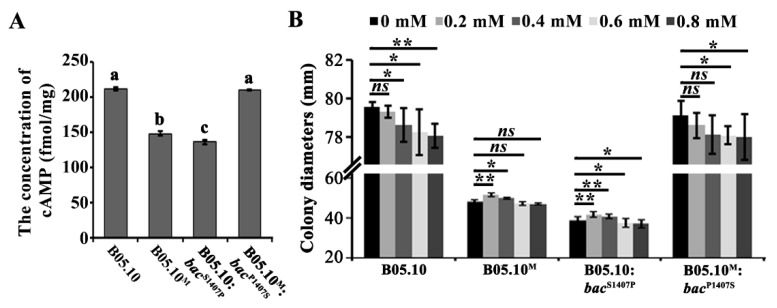
The S1407P mutation in *bac* affected intracellular cyclic AMP (cAMP) contents. (**A**) Intracellular levels of cAMP in the B05.10, B05.10^M^, B05.10:*bac*^S1407P^, and B05.10^M^:*bac*^P1407S^ after incubation on CM for 3 days under light conditions. Different letters on the columns indicate significant differences (*p* < 0.05). (**B**) The colony diameters of the above strains on CM medium with and without exogenous cAMP (the final concentrations were from 0 to 0.8 mM) were measured. Single and double asterisks above the bars indicate significant differences at *p* < 0.05 (*) and *p* < 0.01 (**), respectively; *ns* indicates no significant difference. Student’s *t*-test was used.

**Figure 6 ijms-21-02912-f006:**
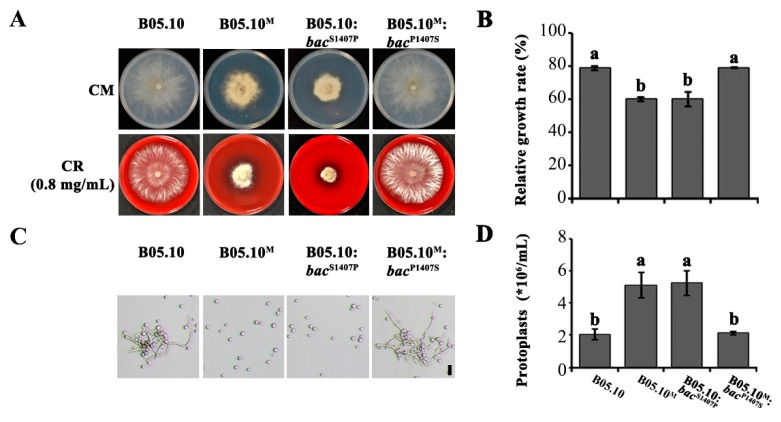
The S1407P mutation in *bac* affected the sensitivity of the fungus to cell wall stress. (**A**) Colony morphology of B05.10, B05.10^M^, B05.10:*bac*^S1407P^, and B05.10^M^:*bac*^P1407S^ were shown after incubation with Congo Red (CR, the final concentration was 0.8 mg/mL) on CM medium for 3 days. (**B**) Relative mycelial growth rates of the strains shown in (**A**). The relative growth rate of each strain was the ratio of colony diameter on the CM medium supplemented with CR to CM medium (control) after incubation for 3 days. (**C**) Generation of protoplasts from mycelia of the above strains treated with cell wall-degrading enzyme for 2 h at 23 °C. Bar = 20 μm. The mycelia were collected from the liquid ME after the conidia shaking 16 h (the initial concentration of conidia was 10^8^ /mL). (**D**) Number of protoplasts of the strains shown in (**C**). Different letters on the columns indicate significant differences (*p* < 0.05).

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
