# Peer review of "A Single Nucleotide Mutation in Adenylate Cyclase Affects Vegetative Growth, Sclerotial Formation and Virulence of Botrytis cinerea"

_ijms, 2020, doi:10.3390/ijms21082912_

Round 1

Reviewer 1 Report

The authors Chen et al. report on the identification of a single nucleotide polymorphism (SNP) in the plant pathogenic fungus Botrytis cinerea affecting growth and virulence. The mutation in BAC (adenylate cyclase) was identified in a derivative of the frequently used recipient strain B05.10 by whole genome sequencing. The mutation-phenotype linkage was confirmed in two ways (smart replacement strategies!), by introducing the point mutation in the wild type B05.10 and introducing the wild-type copy of bac in B05.10M carrying the SNP.

The manuscript is well-written, required controls were performed and most methods were well-described. I think that the manuscript with minor modifications as discussed below is worth to be published though not much new information of the cAMP pathway is presented because it emphasizes a big problem, i.e. random mutations happening to strains that are kept and used as recipients for genetic studies in our laboratories. When these mutations are overlooked, genetic studies may result in wrong observations – though for publishing, complementation of deletion mutants should ensure the correctness.

Unfortunately, the authors did not include a ∆bac mutant into their study. Without a comparative analysis of the found mutated strain (B05.10 w/ SNP) with the deletion strain, it is not possible to say whether the point mutation just alters or abolishes BAC activity. I agree with the authors that the reduced cAMP levels in the SNP-containing strains suggest reduced BAC activity, however, the results obtained with exogenous cAMP on the growth of the mutants is not convincing: I would expect restoration of growth rates almost to wild-type levels!

Minor comments

1) Shorten the introduction by deleting irrelevant information (focus on B. cinerea); i.e.

- “In fungi, cross-talk among these signalling pathways….involved in both signalling pathways [11].”

- In the model fungus Aspergillus nidulans, light responses….expression of the redl-light receptor gene is regulated in phytopathogenic fungi has not been clearly elucidated.”

2) Introduction: not clear/true that B. cinerea produces classical “mycotoxins”…I guess the authors mean the “phytotoxins botrydial and botcinic acid”?!

3) Regarding nomenclature: bac (in small italicized letters) is already defined as gene. Thus, “bac gene” can be shortened to “bac”

4) Change the nomenclature for the strain B05.10M:bacP1407S ---- is it really a mutant strain? because the authors claim the expression of a wild-type of BAC, correct? Might be confusing for the readers.

5) Check whether all important information is given. For example, I could not find the information whether the medium for conidial germination contained a carbon source, which lysing enzyme were used for making protoplasts, which kind of cAMP (permeable?!) and CongoRed etc. was used.

6) Please careful re-consider the interpretation of the cAMP supplementation data show in Fig. 5B (see comment above)

7) Ad Fig. 4: ACs are multi-domain proteins. Please show a scheme of the protein domains of BAC and the site of the mutation. Is it in a domain with a specific function, e.g. in the catalytic domain? The findings should be discussed in the context, i.e. whether the found phenotypes can be due to reduced enzymatic activity and/or misregulation of BAC or something else (altered protein-protein interaction with RAS or AC-associated proteins).

8) Discussion: the authors should delete the sentence “The function of SNP has been frequently reported in animals and plants, but rarely reported in fungi, in…”

Author Response

Point-by-point response to the two reviewers.

Response to Reviewer 1 Comments

The authors Chen et al. report on the identification of a single nucleotide polymorphism (SNP) in the plant pathogenic fungus Botrytis cinerea affecting growth and virulence. The mutation in BAC (adenylate cyclase) was identified in a derivative of the frequently used recipient strain B05.10 by whole genome sequencing. The mutation-phenotype linkage was confirmed in two ways (smart replacement strategies!), by introducing the point mutation in the wild type B05.10 and introducing the wild-type copy of bac in B05.10M carrying the SNP.

The manuscript is well-written, required controls were performed and most methods were well-described. I think that the manuscript with minor modifications as discussed below is worth to be published though not much new information of the cAMP pathway is presented because it emphasizes a big problem, i.e. random mutations happening to strains that are kept and used as recipients for genetic studies in our laboratories. When these mutations are overlooked, genetic studies may result in wrong observations – though for publishing, complementation of deletion mutants should ensure the correctness.

Unfortunately, the authors did not include a ∆bac mutant into their study. Without a comparative analysis of the found mutated strain (B05.10 w/ SNP) with the deletion strain, it is not possible to say whether the point mutation just alters or abolishes BAC activity. I agree with the authors that the reduced cAMP levels in the SNP-containing strains suggest reduced BAC activity, however, the results obtained with exogenous cAMP on the growth of the mutants is not convincing: I would expect restoration of growth rates almost to wild-type levels!

Response: Thanks for your comments. The growth and pathogenicity phenotypes of bac point mutant B05.10:bacS1407P were similar to those of the reported ∆bac mutant. We are trying to knock out the bac gene to obtain the mutant, and hope it will be helpful to comprehensively analyze the mechanism of BAC.

According to the reference (Klimpel, A. et al), the growth rate of colonies was only partially restored by the addition of exogenous cAMP in ∆bac mutant. In addition, in Penicillium digitatum and Aspergillus flavus, exogenous addition of cAMP also only slightly restored the growth rate of adenylate cyclase gene deletion mutants (Yang, K. et al; Wang, W. et al). This may be as a result of adenylate cyclase has other functions besides cAMP production. We discussed this in the revised manuscript.

Minor comments

1) Shorten the introduction by deleting irrelevant information (focus on B. cinerea); i.e.

- “In fungi, cross-talk among these signalling pathways….involved in both signalling pathways [11].”

- In the model fungus Aspergillus nidulans, light responses….expression of the redl-light receptor gene is regulated in phytopathogenic fungi has not been clearly elucidated.”

Response: Thanks for your suggestion. The introduction has been shortened in the revised manuscript.

2) Introduction: not clear/true that B. cinerea produces classical “mycotoxins”…I guess the authors mean the “phytotoxins botrydial and botcinic acid”?!

Response: Thanks for your suggestions and we have changed it in the revised manuscript.

3) Regarding nomenclature: bac (in small italicized letters) is already defined as gene. Thus, “bac gene” can be shortened to “bac”

Response: Thanks for your suggestion. This has been corrected in the revised manuscript.

4) Change the nomenclature for the strain B05.10M:bacP1407S ---- is it really a mutant strain? because the authors claim the expression of a wild-type of BAC, correct? Might be confusing for the readers.

Response: Thanks for your suggestion. This has been changed in the revised manuscript. We completed protoplast transformation in B05.10M, and restored the proline at 1407 to serine to obtain the strain B05.10M:bacP1407S. Therefore, it may be more appropriate to name it bac complementary strain.

5) Check whether all important information is given. For example, I could not find the information whether the medium for conidial germination contained a carbon source, which lysing enzyme were used for making protoplasts, which kind of cAMP (permeable?!) and CongoRed etc. was used.

Response: Thanks for your suggestion. The medium for conidial germination is liquid GB medium (3.05 g Gamborg B5, 3.6 g glucose in 1 L water). The lysing enzyme (Sigma-Aldrich, USA), the cAMP (Macklin, China) and Congo Red (Sangon, China) were used, respectively. We have modified them in the revised manuscript.

6) Please careful re-consider the interpretation of the cAMP supplementation data show in Fig. 5B (see comment above)

Response: Thanks for your comment. In B. cinerea, Penicillium digitatum and Aspergillus flavus, the growth rate of colonies was only slightly restored by the addition of exogenous cAMP in adenylate cyclase gene deletion mutants. We speculate that adenylate cyclase has multiple functional domains, which may have other functions besides cAMP production. We discussed this in the revised manuscript.

7) Ad Fig. 4: ACs are multi-domain proteins. Please show a scheme of the protein domains of BAC and the site of the mutation. Is it in a domain with a specific function, e.g. in the catalytic domain? The findings should be discussed in the context, i.e. whether the found phenotypes can be due to reduced enzymatic activity and/or misregulation of BAC or something else (altered protein-protein interaction with RAS or AC-associated proteins).

Response: Thanks for your suggestion. The typical AC amino acid sequences of plant pathogenic fungi contained G-alpha binding domains, Ras association domains, PP2Cc-type phosphatase domains and AC catalytic enzyme domains. In the study, the mutant amino acid S1407 is conserved in fungi and located in the BAC phosphatase domain. We have added this information in the revised manuscript (Figure S3).

 According to the amino acid characteristics, serine is a polar amino acid, while proline is a nonpolar amino acid. Due to the R group of proline forms a loop with the amino group, it is easy to affect the formation of the α-helix of the secondary structure of the protein, making α-helix turn into a β - fold, which may affect the conformation of protein. We further used the SWISS-3d web to predict the protein structures of two BAC and showed this information in the revised manuscript (Figure S4); it was found that the S1407P mutation protein BACS1407P and wild type BACS1407 have different protein conformations.

Therefore, we speculated that the S1407P mutation may affect the protein conformation of BAC, resulting in decreased enzyme activity, thus affect the development and pathogenicity. We have discussed this in the revised manuscript.

8) Discussion: the authors should delete the sentence “The function of SNP has been frequently reported in animals and plants, but rarely reported in fungi, in…”

Response: Thanks for your suggestion. This has been deleted in the revised manuscript.

Reviewer 2 Report

This study verified a SNP that was discovered through resequencing and structural variance prediction. Mutations in the WT and the mutant confirmed the single nucleotide determinant of the observed phenotypic change.

Some more clarifications can be made on which selective marker was actually inserted in the site. The authors mentioned either hyg or nat was inserted. Or were both transformation was made? Please include the negative control of WT bac gene tagged with only hyg or nat.

Was homokaryons/heterokaryons confirmed by PCR? How was it achieved if they were homokaryons? In resting conidia of B. cinerea strain B05. 10, it was reported to be observed between 2 and 7 nuclei (average, 3.7) in reference to Leroch M, Kleber A, Silva E, et al. Transcriptome profiling of Botrytis cinerea conidial germination reveals upregulation of infection-related genes during the prepenetration stage. Eukaryot Cell. 2013;12(4):614–626. doi:10.1128/EC.00295-12.  Also, some papers suggest that sometimes it can be tricky to obtain homokaryons by isolation of single colonies: https://www.ncbi.nlm.nih.gov/pmc/articles/PMC4010548/ So it would be insightful for the readers to learn how homokaryons were obtained.

Please provide more information on the domain of S1407 in discussion. Is it predicted to affect the folding of bac protein or does the mutation have functional significance affecting catalytic activity. 

Please include raw data in the supplemental data, e.g. colony diameters etc.

Author Response

Point-by-point response to the two reviewers.

Response to Reviewer 2 Comments

This study verified a SNP that was discovered through resequencing and structural variance prediction. Mutations in the WT and the mutant confirmed the single nucleotide determinant of the observed phenotypic change.

Some more clarifications can be made on which selective marker was actually inserted in the site. The authors mentioned either hyg or nat was inserted. Or were both transformation was made? Please include the negative control of WT bac gene tagged with only hyg or nat.

Response: Thanks for your suggestion. I'm sorry I may not have described this part clearly. When mutants were generated via protoplast transformation, hyg or nat could be used as resistance selection markers in the vector. In this study, nat was used as a selective marker to obtain bac point mutants. We performed site-directed mutagenesis of the bac in B05.10 and B05.10M using a homologous recombination strategy, and generated strain B05.10:bacS1407P and B05.10M:bacP1407S, respectively. Through the mutual verification of this bidirectional mutation, the function of S1407P point mutation can be fully proved. This has been corrected in the revised manuscript.

Was homokaryons/heterokaryons confirmed by PCR? How was it achieved if they were homokaryons? In resting conidia of B. cinerea strain B05. 10, it was reported to be observed between 2 and 7 nuclei (average, 3.7) in reference to Leroch M, Kleber A, Silva E, et al. Transcriptome profiling of Botrytis cinerea conidial germination reveals upregulation of infection-related genes during the prepenetration stage. Eukaryot Cell. 2013;12(4):614–626. doi:10.1128/EC.00295-12.  Also, some papers suggest that sometimes it can be tricky to obtain homokaryons by isolation of single colonies: https://www.ncbi.nlm.nih.gov/pmc/articles/PMC4010548/ So it would be insightful for the readers to learn how homokaryons were obtained.

Response: Thanks for your comment. We also agree with your opinion, there are indeed multiple nuclei in the spores and hyphae of Botrytis cinerea, which is also a common problem in the genetic research of Botrytis cinerea. Therefore, it is usually necessary to perform multiple isolation of single spore colony to obtain homokaryons. When the transformants are obtained by protoplast transformation, the accuracy of the upstream and downstream insertion positions of the transformed fragments is first identified by PCR with specific primers (such as primer pairs P7/P8 and P9/P10 in the study). Then, the transformants were subjected to multiple isolation of single spore colony, and were identified by diagnostic PCR of specific fragments and gene sequencing, until homokaryons were obtained. And this is also a commonly used method in the study of Botrytis cinerea.

Please provide more information on the domain of S1407 in discussion. Is it predicted to affect the folding of bac protein or does the mutation have functional significance affecting catalytic activity?

Response: Thanks for your suggestion. The typical AC amino acid sequences of plant pathogenic fungi contained G-alpha binding domains, Ras association domains, PP2Cc-type phosphatase domains and AC catalytic enzyme domains. In the study, the mutant amino acid S1407 is conserved in fungi and located in the BAC phosphatase domain. We have added this information in the revised manuscript (Figure S3).

 According to the amino acid characteristics, serine is a polar amino acid, while proline is a nonpolar amino acid. Due to the R group of proline forms a loop with the amino group, it is easy to affect the formation of the α-helix of the secondary structure of the protein, making α-helix turn into a β - fold, which may affect the conformation of protein. We further used the SWISS-3d web to predict the protein structures of two BAC and showed this information in the revised manuscript (Figure S4); it was found that the S1407P mutation protein BACS1407P and wild type BACS1407 have different protein conformations.

Therefore, we speculated that the S1407P mutation may affect the protein conformation of BAC, resulting in decreased enzyme activity, thus affect the development and pathogenicity. We have discussed this in the revised manuscript.

Please include raw data in the supplemental data, e.g. colony diameters etc.

Response: Thanks for your suggestion. We have added the data to the supplementary data and formatted as a new Table S1.